# *DNAJB1-PRKACA* in HEK293T cells induces *LINC00473* overexpression that depends on PKA signaling

**Stephanie S. Kim**[1☉], **Ina Kycia**[1☉], **Michael Karski**[1☉], **Rosanna K. Ma**[2☉], **Evan A. Bordt**[3], **Julian Kwan**[4], **Anju Karki**[1], **Elle Winter**[1], **Ranan G. Aktas**[1], **Yuxuan Wu**[5,6,7], **Andrew Emili**[4], **Daniel E. Bauer**[5,6,7], **Praveen Sethupathy**[2], **Khashayar Vakili**[1]*

**1** Department of Surgery, Boston Children's Hospital, Boston, MA, United States of America, **2** Department of Biomedical Sciences, College of Veterinary Medicine, Cornell University, Ithaca, NY, United States of America, **3** Department of Pediatrics, Lurie Center for Autism, Massachusetts General Hospital, Harvard Medical School, Boston, MA, United States of America, **4** Department of Biochemistry, Center for Networks Systems Biology, Boston University School of Medicine, Boston, MA, United States of America, **5** Division of Hematology/Oncology, Boston Children's Hospital, Boston, MA, United States of America, **6** Department of Pediatric Oncology, Dana-Farber Cancer Institute, Harvard Stem Cell Institute, Broad Institute, Boston, MA, United States of America, **7** Department of Pediatrics, Harvard Medical School, Boston, MA, United States of America

☉ These authors contributed equally to this work.
* khashayar.vakili@childrens.harvard.edu

**Data Availability Statement:** All relevant data are within the paper and its Supporting Information file.

## Abstract

Fibrolamellar carcinoma (FLC) is a primary liver cancer that most commonly arises in adolescents and young adults in a background of normal liver tissue and has a poor prognosis due to lack of effective chemotherapeutic agents. The *DNAJB1-PRKACA* gene fusion (DP) has been reported in the majority of FLC tumors; however, its onco-genic mechanisms remain unclear. Given the paucity of cellular models, in particular FLC tumor cell lines, we hypothesized that engineering the DP fusion gene in HEK293T cells would provide insight into the cellular effects of the fusion gene. We used CRISPR/Cas9 to engineer HEK293T clones expressing DP fusion gene (HEK-DP) and performed transcriptomic, proteomic, and mitochondrial studies to characterize this cellular model. Proteomic analysis of DP interacting partners identified mitochondrial proteins as well as proteins in other subcellular compartments. HEK-DP cells demonstrated significantly elevated mitochondrial fission, which suggests a role for DP in altering mitochondrial dynamics. Transcriptomic analysis of HEK-DP cells revealed a significant increase in *LINC00473* expression, similar to what has been observed in primary FLC samples. *LINC00473* overexpression was reversible with siRNA targeting of *PRKACA* as well as pharmacologic targeting of PKA and Hsp40 in HEK-DP cells. Therefore, our model suggests that *LINC00473* is a candidate marker for DP activity.

**Funding:** This work was supported by the Tisch Families' Faculty Development Fund and CHMC Surgical Foundation, Inc. The funders had no role in study design, data collection and analysis, decision to publish, or preparation of the manuscript.

**Competing interests:** The authors have declared that no competing interests exist.

## Introduction

Fibrolamellar carcinoma (FLC) is a primary liver cancer that most commonly arises in healthy adolescents and young adults in a background of normal liver tissue [1–3]. The only chance of cure is surgical resection. A large study from the Childhood Liver Tumour Strategy Group (SIOPEL), published in 2013, demonstrated that patients with FLC had a 5-year event-free survival of ~18% and an overall survival of 15% [4]. The dismal prognosis of FLC is due in large part to the lack of effective chemotherapeutic agents, creating an area of urgent and unmet need in the field.

In 2014, the *DNAJB1-PRKACA* (DP) gene fusion was initially described in a cohort of FLC patients [5]. Subsequent studies have demonstrated the presence of this fusion gene in the majority of FLC tumors [6]. The DP gene fusion is the result of a ~400 kb heterozygous deletion in chromosome 19 that results in the fusion of exon 1 of *DNAJB1* to exons 2–10 of *PRKACA* in most FLC cases [5]. The fusion gene remains under the control of the *DNAJB1* promoter. *DNAJB1* gene codes for a heat shock protein (HSP40) and *PRKACA* gene codes for the Cα catalytic subunit of protein kinase A (PKA-Cα). PKA is a cyclic AMP (cAMP)-dependent protein kinase with its holoenzyme consisting of a tetramer of two regulatory (R) subunits and two catalytic (C) subunits that contain the active kinase site. An increase in the cytosolic second messenger cAMP results in binding to the regulatory subunits, with subsequent dissociation and activation of the catalytic subunits of PKA. The free catalytic subunits subsequently phosphorylate proteins involved in a number of cellular processes such as metabolism, synaptic transmission, differentiation, growth, and development [7–9].

Although the mechanisms through which the fusion gene leads to FLC are unclear, the oncogenic potential of *DNAJB1-PRKACA* fusion has been confirmed in mouse models by introducing the fusion gene into the liver using cDNA with a sleeping beauty transposon or using CRISPR/Cas9 engineering [10,11]. In these models, the mice developed FLC tumors after a period of nearly 1.5 years. These studies were important in demonstrating the oncogenic potential of the fusion gene. However, there are some challenges with using these animal models for therapeutic screening due to the latency of tumor development as well as the inability to perform high-throughput drug screening. In addition to these models, at least one patient-derived xenograft (PDX) model of FLC [12], as well as cell lines derived from this PDX [13–15], have been established. Despite the notable advances that this PDX has facilitated, it harbors some practical limitations, including that it is very slow growing. To address this limitation, some cell line models have been developed, including lentiviral over-expression of DP in liver cancer lines such as HepG2 [6,13] and CRISPR/Cas9-based deletion to create the *DNAJB1-PRKACA* fusion in AML12 cells [13,16]. An important limitation of the former is that the cells are derived from a different type of cancer (hepatoblastoma) and salient limitations of the latter are that the cells are murine in origin and harbor the human transforming growth factor-alpha (TGF-α) transgene.

We hypothesized that engineering the *DNAJB1-PRKACA* fusion gene in a HEK293T cell line would provide insight into the potential oncogenic mechanism of the fusion gene. Even though HEK293T cells are not of liver origin, we decided to use them due to the fact they are easy to handle in culture and are derived from human embryonic cells, which is relevant since FLC tumors exhibit molecular profiles that resemble that of early stage progenitor cells [12]. We hypothesized that even the presence of the fusion protein in HEK293T cells may still reveal key protein-protein interactions and highlight critical downstream signaling molecules and pathways. We elected not to use available liver cancer cell lines since they contain numerous other oncogenic alleles, which could confound our results. In this report, we characterize the transcriptomic, proteomic, and mitochondrial phenotypes of the DP engineered HEK293T cells (HEK-DP) and present *LINC00473* as a marker for DP activity.

## Results

### Engineering DNAJB1-PRKACA fusion gene in HEK293T cells

In order to create the *DNAJB1-PRKACA* fusion gene, we used CRISPR/Cas9 editing to delete the approximately 400 kb segment of DNA in HEK293T cells (HEK-WT) as observed in FLC tumors. To our knowledge, there is only one other cell line, a mouse hepatocyte cell line (AML12) [16], which has been engineered in a similar fashion. We identified 12 HEK293T clones (A1, A3, A5, A6, A9, A11, B2, B6, C2, C6, C10, C11) which contained the *PRKACA-DNAJB1* fusion gene (HEK-DP) as determined by PCR analysis across the expected junction site (Figs 1 and S1). The use of the same primers in HEK-WT did not result in a PCR product. The PCR product for each clone was then subjected to DNA sequencing which demonstrated 90–100% match to the expected DNA sequence across the junction site (Fig 1C). The expression of the DNAJB1-PRKACA fusion protein (DP) was confirmed using Western blot analysis (Fig 1D). The wild-type PKA-Cα is demonstrated as a 41kD band and DP as a 46kD band. All 12 clones expressed the fusion gene and several clones demonstrated a higher expression of DP protein relative to the wild-type PKA-Cα protein (A11, C6, C11). Patient-derived FLC tumor sample was used as a positive control (Fig 1D). We selected clones A9 and A11 for further analysis given that A11 demonstrated a higher expression of DP protein compared to A9. All methods were carried out in accordance with relevant guidelines and regulations. All experimental protocols were approved by the Institutional Review Board and/or the Institutional Biosafety Committee at Boston Children's Hospital.

### HEK-DP clones share several gene expression aberrancies with FLC tumors

Transcriptome analysis using a human gene chip array was performed on A9, A11, and HEK-WT cells (Fig 2). Heat maps in Fig 2 represent both upregulated and downregulated

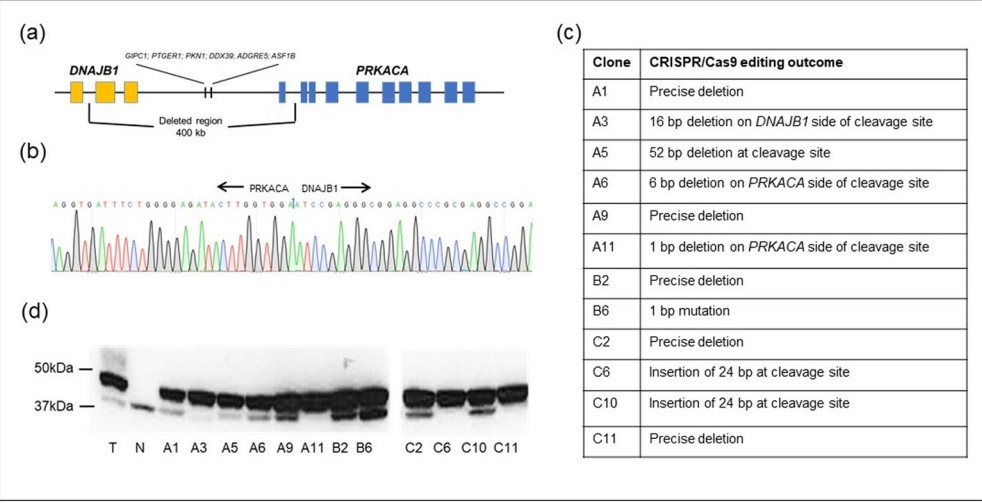

**Fig 1. HEK-DP engineering.** (a) Schematic of chromosome 19 400 kb pair deletion with (b) example of sequencing at the junction site between *DNAJB1* and *PRKACA*. (c) Comparison of DNA sequence at junction site compared to expected sequence of 12 HEK-DP clones. (d) PKA-Cα immunoblot analysis demonstrated DP fusion protein expression by a 46kD band and wild-type PKA-Cα by a 41kD band. Patient FLC tumor sample (T) and normal liver tissue (N) were used as positive and negative controls, respectively. All colonies (A1-H6) demonstrate expression of DP protein with some clones demonstrating a significantly lower level of wild-type PKA-Cα protein expression (A3, A11, C6, C11). Full-length blots are presented in S1 Fig.

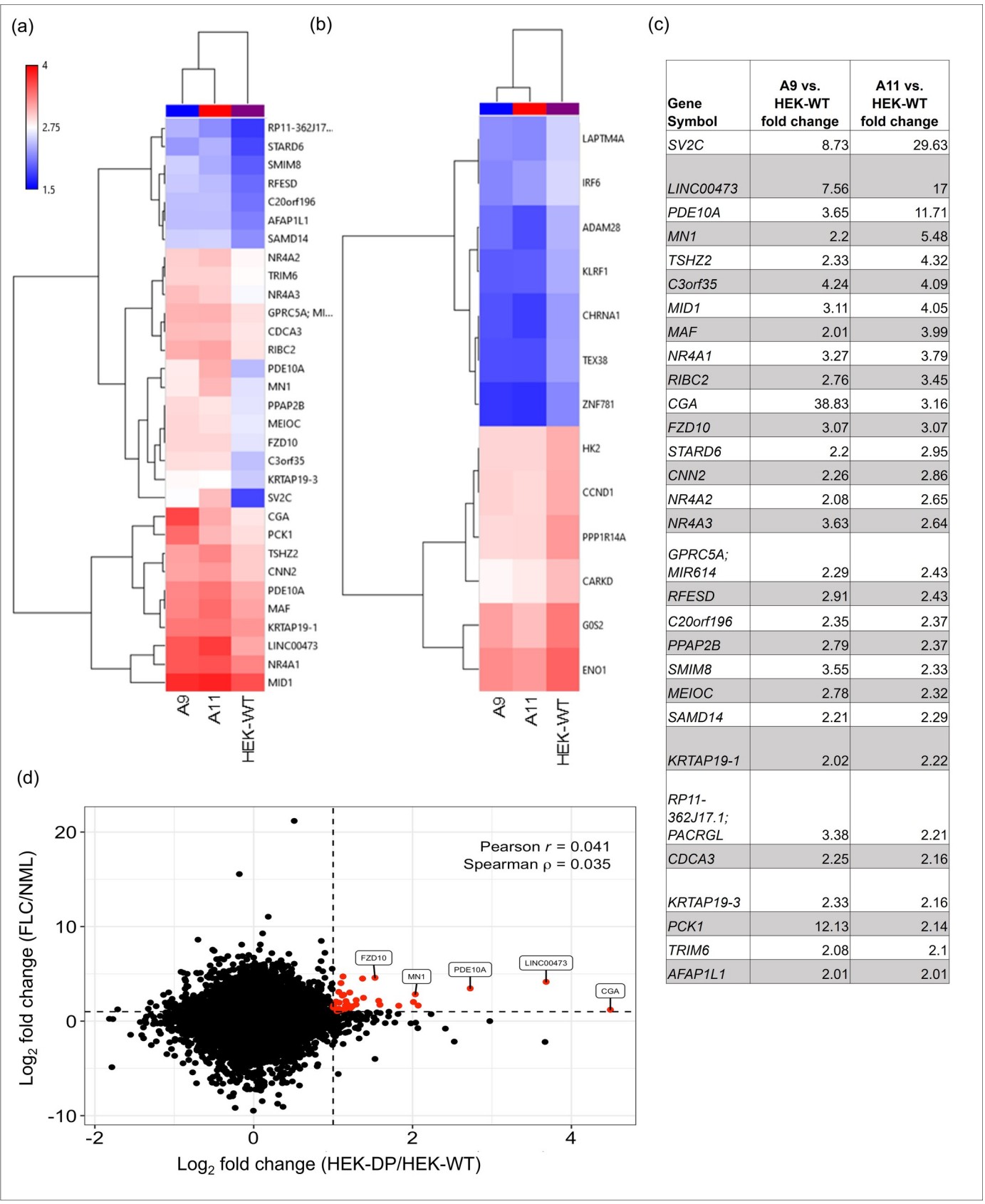

**Fig 2. Clones with DP fusion protein share common alterations with FLC tumors.** Hierarchical clustering of A9, A11 clones at 2 fold transcript change compared to HEK-WT with the most upregulated genes in (a and c) and most downregulated genes in (b). (d) Scatterplot heat map identifies *CGA, LINC00473, PDE10A, MN1, and FZD10* as common upregulated genes in A9 and A11 and a cohort of FLC tumor samples.

genes with >2-fold change in expression. The transcriptome data from A9 and A11 were then compared to 23 primary or metastatic FLC tumor samples and 4 non-malignant liver samples. Our analysis demonstrated similar pattern of increased expression in *CGA, LINC00473, PDE10A, MN1, FZD10* in both HEK-DP cells and FLC samples (Fig 2D). In fact, *LINC00473* and *FZD10* were identified previously as being associated with FLC-specific super enhancers [13], which are known to mark genes that are critical for tumor cell behavior [17]. Subsequent quantitative analysis of the gene chip findings using RT-qPCR confirmed the significant increase in *LINC00473 TV1 and TV2* compared to HEK-WT (Fig 3). Increased expression of *LINC00473*, a long non-coding RNA (lncRNA), has been demonstrated in FLC [13] as well as other cancer types [18,19]. HEK-CT clones which underwent transfection with vectors lacking guide RNA sequences did not demonstrated any differences in expression of either transcript variants of *LINC00473* compared to HEK-WT cells.

## RNAi-mediated suppression of PKA decreases LINC00473 expression in HEK-DP cells

PKA-Cα and DP protein levels were significantly decreased after 48 and 72 hours of treatment with *PRKACA* siRNA (Fig 4A). In order to determine whether the increased expression of *LINC00473* was directly related to PKA-Cα, we assessed their expression in response to *PRKACA* siRNA knockdown following 24, 48, and 72 hours of treatment. There was a significant decrease in the expression of both *LINC00473 TV1* and *TV2* (Fig 4C and 4D) following treatment with *PRKACA*-siRNA (Unpaired t-test, n = 3, p<0.001). These results demonstrate that the expression of *LINC00473* can be modulated by targeting *PRKACA* transcript.

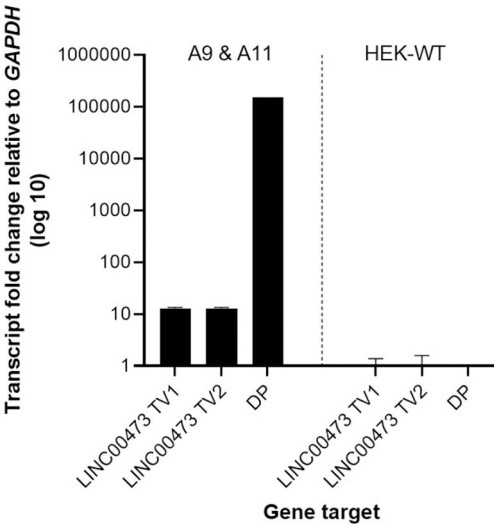

**Fig 3. Expression of *LINC00473* is upregulated in HEK-DP clones.** (a) The two known transcript variants (*TV1* and *TV2*) of *LINC00473* are significantly upregulated in A9 and A11 clones. The expression of the *DNAJB1-PRKACA* fusion transcript (DP) is demonstrated in A9 and A11.

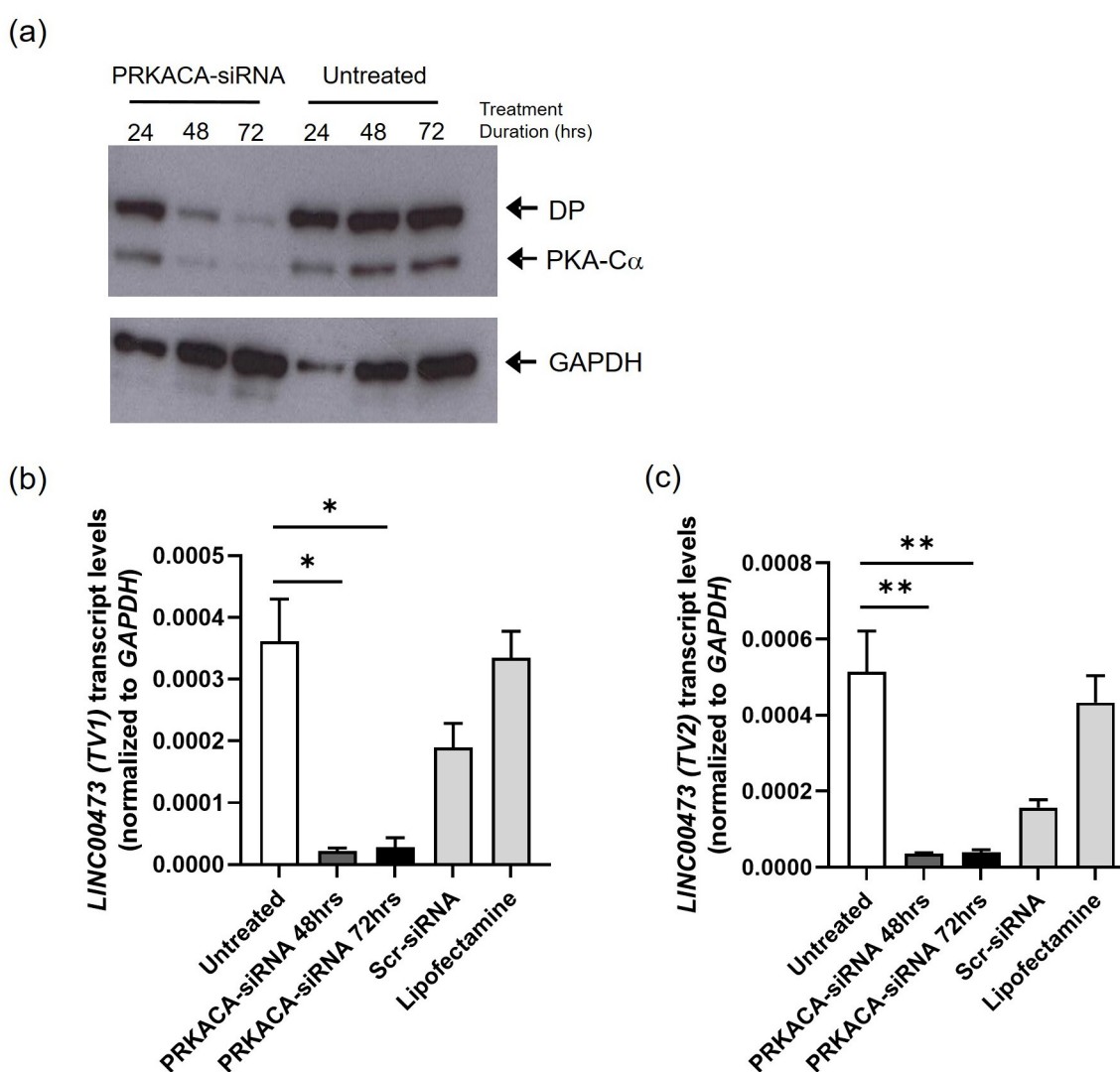

**Fig 4. siRNA targeting *PRKACA* transcript decreases expression of both transcript variants of *LINC00473* in A9 clone.** (a) siRNA directed at *PRKACA* effectively decreases both PRKACA and DP protein levels following 48 and 72 hours of treatment. Full-length blots are presented in S1 Fig. (b-c) Treatment with *PRKACA*-siRNA demonstrates statistically significant decrease in *LINC00473 TV1* and *TV2* transcript levels at 48 and 72 hours. (*-p<0.005; **-p<0.05).

## Pharmacologic targeting of PKA or Hsp40 decreases LINC00473 expression in HEK-DP cells

Treatment of A9 and A11 with known PKA inhibitor H89 [20] for 24 hours resulted in a significant decrease in *LINC00473 TV1* expression (Fig 5A) with a higher IC50 for A11 compared to A9. H89 also resulted in a decrease in *LINC00473 TV2* expression in A9 and A11, however, at a higher dose compared to *TV1* (Fig 5). In order to determine the effects of targeting Hsp40 component of DP on *LINC00473*, we tested KNK437. KNK437 is a benzylidene lactam compound, which has been shown to inhibit the induction of Hsp40, Hsp70, and Hsp105 [21]. KNK437 treatment resulted in a statistically significant downregulation of *LINC00473 TV1 and TV2* in both A9 and A11 cells (Fig 5C and 5D).

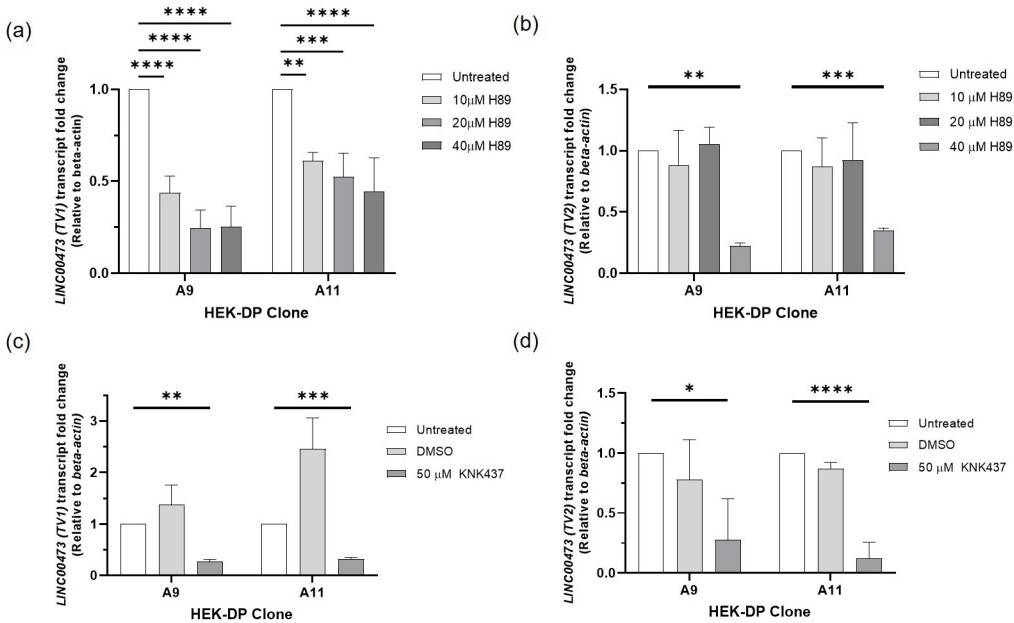

**Fig 5. Pharmacologic targeting of PKA with H89 and Hsp40 with KNK437 results in downregulation of *LINC00473* in HEK-DP cells.** (a) H89 treatment of A9 and A11 clones demonstrates decreased expression of *LINC00473 (TV1)*; (b) H89 treatment of A9 and A11 clones demonstrates decreased expression of *LINC00473 (TV2)*; (c-d) KNK437 treatment results in decreased expression of *LINC00473 TV1 and TV2*. (*p<0.05; **p<0.01; ***p<0.001; ****p<0.0001).

## A11 clone demonstrates increased proliferation rate compared to HEK-WT

In order to examine whether the expression of DP has an impact on the proliferation rate of the HEK-DP cells, we assessed the proliferation rates of A9, A11, and HEK-WT over a 6-day time course (Fig 6). Comparison of the cell count on each day revealed a significant difference

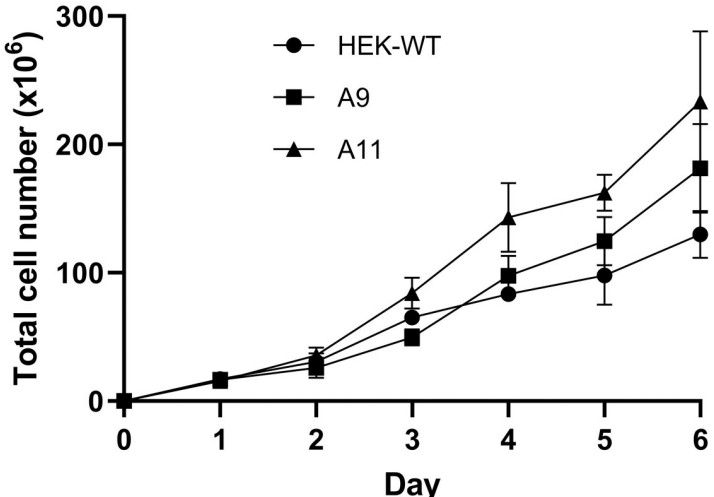

**Fig 6. A11 clone has a significantly higher proliferation rate compared to HEK-WT.** A11 has a higher proliferation rate compared to HEK-WT starting at day 3 (p<0.05). There is no statistically significant difference in rate of proliferation between A9 and HEK-WT.

between A11 and HEK-WT starting on day 3 (Student t-test, p<0.05). There was no significant difference between A9 and HEK-WT. An area-under-the-curve (AUC) analysis using an integrated function of best fit (polynomial = 2) also demonstrated higher proliferation rate of A11 compared to HEK-WT (p = 0.003) and A9 (p = 0.004). An alternate AUC analysis using a geometric approach comparing the area of the trapezoid between each day also demonstrated a higher proliferation rate of A11 compared to HEK-WT (p = 0.003) and A9 (0.012). The AUC analyses did not demonstrate a significant difference between A9 and HEK-WT.

## Proteomic analysis of HEK-DP cell lines

In order to identify protein-protein interactions with DP in A9 and A11 clones, we used PKA-Cα antibody for immunoprecipitation (IP) assays given the lack of DP-specific antibodies. The complexes were then subjected to mass spectroscopic analysis. We used PKA-Cα IP in HEK-WT cells as control. Twenty-eight potential DP interacting proteins were identified. Fig 7A heat map demonstrates the proteins identified in response to PKA-Cα immunoprecipitation in A9 and A11 that were not immunoprecipitated in HEK-WT cells. A list of interacting proteins are provided in S1 Table and demonstrates interaction between DP and proteins known to be associated with various compartments including cytoplasm, nucleus, mitochondria, and endoplasmic reticulum. Gene ontology analysis comparing HEK-DP and HEK-WT cells identified pathways in protein metabolism (R-HAS-392499) and RNA binding (S2 Table). Although the analysis revealed a number of common binding partners between A9 and A11 cells, principal component analysis also revealed groups of proteins that were specific to each cell line (Fig 7B–7D). DNAJB1 is identified as one of the preferentially immunoprecipitated proteins which reflects the DNAJB1 segment of the fusion protein and provides support for the accuracy of the proteomic analysis.

Currently, there are no studies which have performed a wide proteomic analysis to identify interactions between DP and other proteins. A study by Turnham et al. [16] suggested an interaction between DP and HSP70 in the AML12 engineered mouse cell line. We assessed whether this interaction is also present in the HEK-DP cell lines using co-immunoprecipitation Western blot analysis. Our results demonstrated that Hsp70 interacts with PKA-Cα in A9 and A11 cells (S3 Fig) confirming the previous findings. Based on our proteomic analysis, we performed co-immunoprecipitation and Western blot analysis directed the interaction of PKA and BAG2. Our results demonstrate co-precipitation of PKA and DP with BAG2 (S4 Fig).

## HEK-DP cell lines demonstrate alterations in mitochondrial morphology

One of the characteristic histologic features of FLC is the eosinophilic nature of the cells with significant granular cytoplasm attributed to abundant mitochondria [22]. Therefore, we sought to examine whether the expression of DP in A9 and A11 clones were associated with any mitochondrial alterations. We used the mitochondrial-specific Tomm20 immunostaining to assess mitochondrial morphology in the HEK-DP and HEK-WT cells (Fig 8A). Mitochondria are known to exist in a dynamic network in which they join through the process of fusion and divide through fission. An imbalance in this dynamic process will influence critical cellular processes including generation of ATP, apoptosis, cell cycle regulation, mitophagy, in addition to other processes. Disorders in mitochondrial fission and fusion have been reported in cancers, cardiovascular disease, and neurodegenerative diseases [23]. There was a significant increase in the frequency of smaller mitochondria (<1 μ m in length) in A9 and A11 clones compared to HEK-WT cells (Kruskal-Wallis test, p<0.0001) in addition to a significant decrease in the frequency of mitochondria 1–3μ m in length in A9 and A11 cells compared to HEK-WT (2-way ANOVA, p<0.05) (Fig 8B and 8C). These findings correlated with a higher

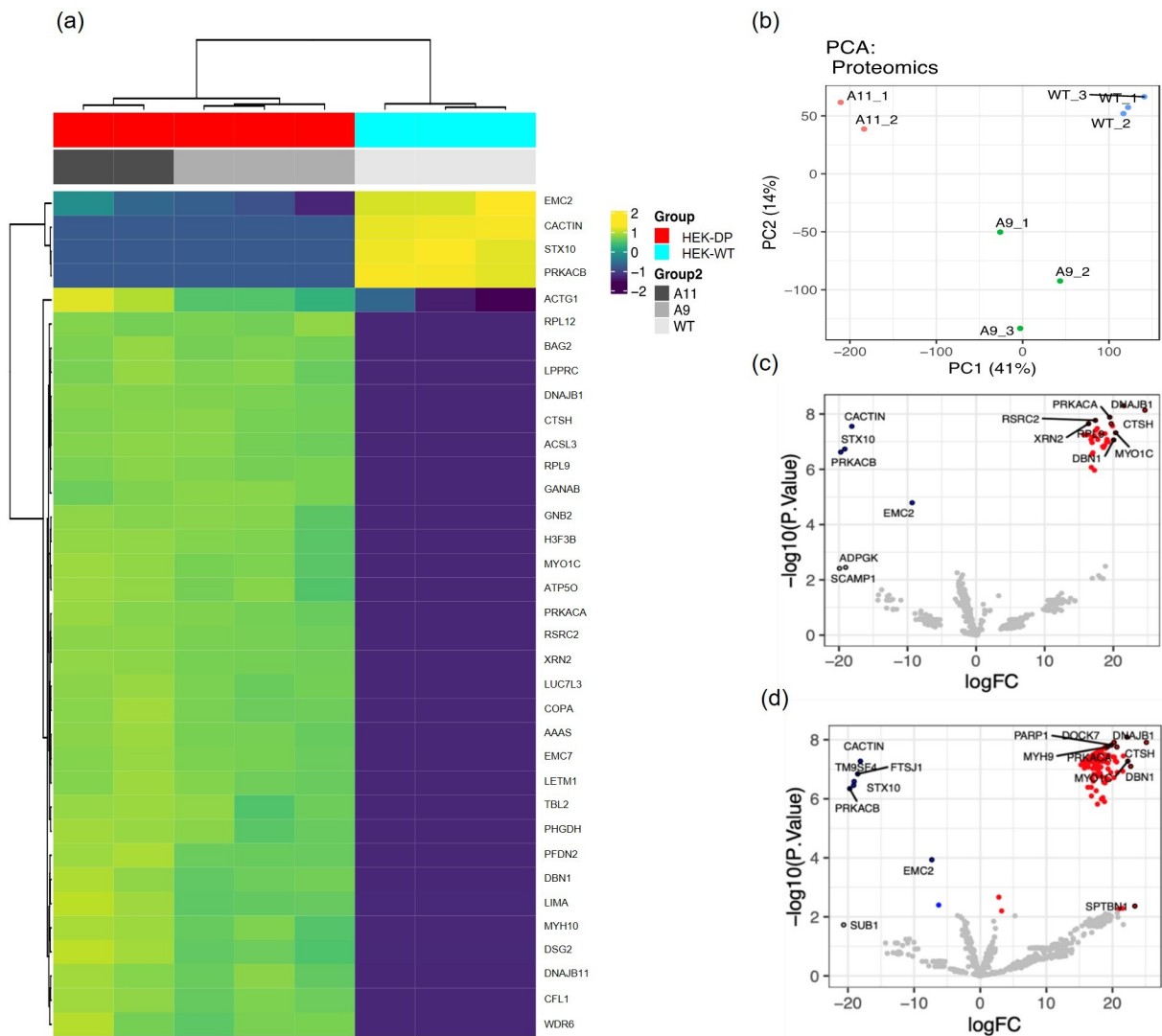

**Fig 7. Proteomic analysis reveals potential DP interacting proteins.** Immunoprecipitation of A11 and A9 protein homogenates with PKA-Cα antibody identifies a set of proteins which are not present in HEK-WT immunoprecipitates and thus attributed to the presence of DP in these clones. Heat map (a) and principal component analysis (b) demonstrate similarities between A9 and A11 as well as tight grouping of the replicate analyses, respectively. Volcano plots highlight potential DP interacting proteins (red) in A9 vs HEK-WT (c) and A11 vs HEK-WT (d).

frequency of mitochondria with smaller volumes in A11 (p<0.0001) and A9 (p<0.05) compared to HEK-WT cells (Fig 8D and 8E). Further analysis also revealed a decrease in the number of branches, large networks, and overall networks in A9 and A11 clones compared to HEK-WT cells (One-way ANOVA, p<0.0001) (Fig 8F–8H). The mitochondrial morphologic alterations seen in A9 and A11 are suggestive of increased fission.

## Discussion

There is an urgent and unmet need for effective therapies for FLC. Given that FLC tumor cells are difficult to grow and maintain in culture (author's personal experience), developing novel in vitro models in order to elucidate the cellular mechanisms of DP oncogene and test

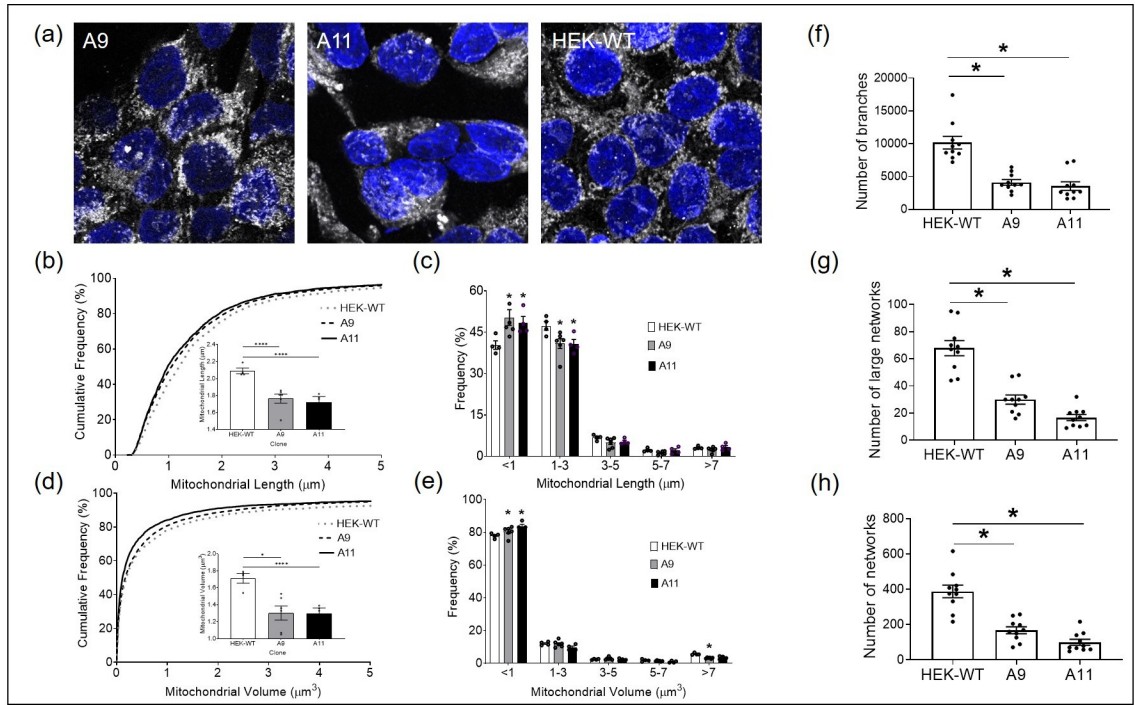

**Fig 8. HEK-DP clones demonstrate increased mitochondrial fission.** Mitochondrial morphology is assessed using Tomm20 immunostaining (a) in A9, A11, and HEK-WT cells. The frequency of mitochondria under 1μm in length is significantly increased in A9 and A11 clones (b and c). Mitochondrial volume is significantly decreased in A9 and A11 compared to HEK-WT (d and e). In addition, the number of branches, large networks, and overall networks are significantly decreased in A9 and A11 (f-h) (*- p<0.05; **** -p<0.0001).

therapeutics is critical. Here we describe a newly engineered HEK293T cellular model of FLC expressing the DP oncogene. Since HEK293T cells are easy to maintain and manipulate, we leveraged them to refine our gene editing approach that we eventually intended for liver progenitor cells. Following successful editing of the DP fusion gene, we noted some interesting findings that resemble FLC and therefore proceeded with further characterization of the HEK-DP cell lines. Our findings suggest that DP results in increased expression of *LINC00473* similar to FLC. We also identified 27 proteins that potentially interact with DP. In addition, we observed increased mitochondrial fission in the HEK-DP cells. We acknowledge that the HEK293T cells are clearly different from the likely cells of origin in FLC, though these have not yet been defined. Despite this difference, we hypothesize that the expression of DP in this cell type may still be of scientific value and contribute to developing a better understanding of DP cellular function in the study of a cancer with a small and limited number of current models.

We initially developed 12 HEK-DP clones and subsequently selected the A11 and A9 clones for further in-depth analysis as both had 100% of the expected sequence at the fusion site. In addition, A11 had a significantly higher expression of the DP fusion protein with no expression of wild-type PKA-Cα compared to the A9 clone. We hypothesized that this difference in protein expression would result in molecular and physiologic differences between these two clones, which would assist in delineating some downstream cellular effect of DP. This is partly evident in the higher proliferation rate of A11 compared to A9.

The key finding in the HEK-DP clones was the increased expression of *LINC00473* transcript, which is also observed in FLC tumors (Fig 4D) and has been previously reported [24].

*LINC00473* is an intergenic transcript, which belongs to the long noncoding RNA (lncRNA) family: RNAs of size >200 nucleotides that are not translated into proteins. lncRNAs have been shown to play a role in chromosome architecture, chromosomal interactions, chromatin remodeling, nuclear bodies regulation, mRNA turnover, and regulation of translation [25]. In addition, there are a number of cancer-associated lncRNAs that have been identified and proposed as markers for cancer diagnosis, prognosis, and prediction of therapeutic responsiveness [26]. Specifically, *LINC00473* has been studied in the context of a number of cancers including lung [18], liver [27], colon [28], and pancreas [29] in addition to others. In these studies, *LINC00473* has been implicated in tumor survival, increased invasiveness, tumorigenesis, and chemotherapy resistance through a number of different mechanisms. Our observation of increased *LINC00473* expression in the majority of HEK-DP clones, matching what has been shown before in HepG2 cells [13], and the significant suppression of its expression with PKA-Cα siRNA provides further important evidence that increases in *LINC00473* in FLC is due to the DP fusion gene. The siRNA experiments clearly demonstrated a consistent and significant downregulation of *LINC00473* in response to treatment with PRKACA-siRNA. In addition, chemical inhibition of PKA in both A9 and A11 clones with H89 demonstrated a downregulation of transcript variants of *LINC00473*. The IC50 in response to H89 treatment was different between TV1 and TV2, suggesting potential differences in the sensitivity of these transcript in response to PKA targeting. Interestingly, targeting DP through KNK437, an Hsp40 inhibitor, also resulted in a decrease in *LINC00473* expression. Together, our siRNA and pharmacologic inhibitory experiments strongly suggest the reversible nature of *LINC00473* expression and its direct link to DP activity.

A previous study had shown upregulation of *C6orf176* (now recognized as *LINC00473*) in response to activation of cAMP signaling pathway following treatment of human ciliary smooth muscle cells with prostaglandin E2 receptor agonists [30]. A follow-up study [31] by the same group proposed that *C6orf176/LINC00473* is a regulator of cAMP-mediated gene expression given that knockdown of *C6orf176/LINC00473* by siRNA increased the expression of several cAMP signaling-responsive genes (*NR4A3*, *NR4A1*, *AVPI1*, *CRISPLD2*, *CHST6*, *C13orf33*). These findings suggest a negative regulatory effect of *C6orf176/LINC00473* on the expression of these genes. In contrast, we noted an upregulation of *NR4A1*, *NR4A3*, and *CRISPLD2* in the HEK-DP cells suggesting activation of the cAMP signaling in our model. cAMP interacts with the regulatory subunits of PKA-Cα resulting in the release of the PKA-Cα which subsequently phosphorylates downstream targets. A previous study has suggested that the activity of the mutant PKA catalytic subunit in DP fusion is dependent on intracellular cAMP levels [32]. However, the persistence of *LINC00473* expression over several generations in the HEK-DP clones point to an autonomous activation of at least part of the cAMP pathway in our model. *LINC00473* has 2 annotated transcript variants, both of which were overexpressed in the HEK-DP cells and their expression was decreased in response to *PRKACA*-siRNA and pharmacologic targeting of PKA and Hsp40. Whether the upregulation of *LINC00473* in the HEK-DP cells as well as the FLC tumors is a positive regulator of further downstream signaling or simply involved in a negative feedback loop remains unclear at this time. However, as mentioned above, since studies in other cancers suggest that *LINC00473* may be a positive regulator in the cAMP-PKA axis, this will require additional future studies.

Currently, there are no large-scale studies on the proteomics of FLC. We therefore sought to identify binding partners of DP to identify potential protein-protein interactions, which may also contribute to FLC pathology. We identified 28 potential proteins that may interact with DP in several different subcellular locations. There was significant concordance between the two HEK-DP cell lines (A9 and A11) in comparison to HEK-WT cells in identifying potential interactors with DP protein. Proteomic enrichment analysis identified pathways involved in protein

metabolism, double stranded RNA binding, tRNA binding, and mRNA binding. There is a study in which the authors engineered the DP fusion gene in AML12 murine hepatocytes and reported a significant interaction between the DP protein and Heat Shock Protein Family A Member 1A (Hsp70) coded by *HSPA1A* gene. The authors propose that the chaperone-binding domain of DP fusion protein recruits Hsp70 and results in a chaperonopathy leading to activation or enhancement of signaling pathways which promote disease progression [16]. In their model, they propose the interaction of DP fusion protein and Hsp70 results in the activation of RAF--MEK-ERK signaling pathway. In our HEK-DP model, we did not identify an interaction between DP and Hsp70 through the proteomic analysis, however, we did identify it on co-immunoprecipitation studies (S3 Fig). This interaction is presumed to be between the J-domain of the DNAJB1 component of the DP protein and Hsp70. DNAJB1 belongs to the Hsp40 family which are known to bind and regulate the function of Hsp70 proteins as molecular chaperones [33,34]. It is unclear why the proteomic analysis did not identify Hsp70 as a potential interacting protein with DP in contrast to the targeted co-immunoprecipitation studies. This may be due to methodological and technical differences between the two approaches. Interestingly, the mass spectroscopic proteomic data and co-immunoprecipitation studies identified BAG2 protein as an interacting protein with DP, which may be significant since BAG2 is also known to interact with Hsp70. BAG2, belongs to the BAG (BCL-2 associated athanogene 1) family of proteins which were initially identified for their ability to protect cells when exposed to pro-death stimuli [35]. Furthermore, BAG2 has co-chaperone activity with the Hsp70 family of molecular chaperones [36,37]. BAG proteins possess a variable domain which interacts with the ATP binding site of Hsp70 acting as a nucleotide exchange factor. Recently, BAG proteins have gained attention for their potential role in various cancers through modulation of autophagy, epithelial-to-mesenchymal transition, apoptosis, and cytoskeletal re-organization [38,39]. The HEK-DP cells may provide a suitable model for further investigation into the interaction of DP, BAG2, and Hsp70.

Another key finding of our work is the increased mitochondrial fission in HEK-DP cells. These findings clearly suggest an alteration in the mitochondrial dynamics in the HEK-DP cells, which we presume to be due to DP. PKA is known to localize to the mitochondria through binding of its regulatory subunits to A kinase anchoring protein 1 (AKAP1) [40,41]. cAMP/PKA signaling has been shown to modulate the oxidative phosphorylation process in the mitochondria through direct phosphorylation of subunits in complex I, IV, and V of the electron transport chain which is the major source of ATP production [42]. In fact, proteomic analysis of HEK-DP cells revealed potential interaction of DP fusion protein with several mitochondrial proteins including ATP synthase subunit O, long-chain-fatty-acid-CoA ligase 3, leucine-rich pentatricopeptide repeat containing, and LETM1 and EF-hand domain-containing protein 1. LETM1 has been shown to play a significant role in mitochondrial morphology, potassium and calcium homeostasis, and respiratory chain biogenesis [43]. Given the abundant mitochondria in FLC tumor cells [22] and increased fission in our HEK-DP models, further work is required to determine the exact mechanism of how the DP protein results in mitochondrial fission and its implications in the biology of FLC.

One of the limitations of our cell model is that it is not a suitable model to study the oncogenic mechanism of DP in liver cells since the HEK293 cells are not of the same origin and at baseline have a different gene expression pattern. One the other hand, establishment of in vitro FLC cultures as well as patient-derived xenograft models have been challenging and therefore have yet to be a reliable approach for the development of a therapeutic approach for DP-associated FLC. Therefore, our engineered HEK-DP cell lines may have the potential to provide a platform for understanding the regulatory link between DP and *LINC00473* expression, which is present in FLC tumors, and provide the field with a cellular system for screening DP-specific inhibitors by using *LINC00473* as candidate marker for DP activity.

## Materials and methods

### Engineering *DNAJB1-PRKACA* fusion gene (*DP*) into HEK293T cells

The gRNAs specific to *DNAJB1* and *PRKACA* target sites were designed using CHOPCHOP web tool and cloned into sgRNA plasmid PX458 containing Cas9. *PRKACA* target sites were cloned into px458-GFP vector and *DNAJB1* target sites were cloned into px458-mCherry vector to allow for double screening. *PRKACA* guide RNA sequence was 5'-GGGAGATACTTGGTG GAAAG-3' and *DNAJB1* guide RNA sequence was 5'-GGCCTCCGCCCTCGGATTGG-3'. Constructs were transfected into HEK293T cell line using Lipofectamine 3000. GFP and mCherry positive cells were sorted using fluorescence-activated cell sorting (FACS) as single cells and subsequently expanded. The px458-mcherry vector was a kind gift from Dr. Michael Stitzel. Growing cells were selected and screened for the fusion transcript by PCR (Forward: CTGCAATCACAACCCCTCGC; Reverse: GAACAGCCAAATCCCAAGGC) and gel electrophoresis. Positive PCR products were then subjected to DNA sequencing. We also established 5 control clones (HEK-CT) following transfection of HEK293T cells with px458-GFP vectors and px458-mCherry vectors which did not contain the sgRNA sequences. These cells underwent FACS and single-cell colonies were established.

### RNA extraction, cDNA synthesis, and qPCR

RNA was extracted from cells using TRIzol reagent (Life Technologies) according to the manufacturer's instructions. RNA concentrations were measured using spectrophotometry (Nanodrop). 0.5ug of RNA was used for cDNA synthesis using iScript-Reverse transcription Supermix (Biorad Laboratories Inc.) according to the manufacturer's instructions. The cDNA was amplified for *LINC00473* and *GAPDH* mRNAs. GAPDH was used as a housekeeping gene. The primer sequences used for *GAPDH* were 5'-GCTCATTTCCTGGTATGACAACG-3' (forward) and 5'-TGGTCCAGGGGTCTTACTCC-3' (reverse); for *LINC00473-TV1* were 5'-GCA GATATGCGCGTCAGCA-3' (forward) and 5'-GTGCCTCCCTGTGAATTCTCTC-3' (reverse); for *LINC00473-TV2* were 5'-AAATGAAGCGGAAAGCAGATATGC-3' (forward) and 5'-CAACGAG CACCAGAGAATACTAGTG-3' (reverse). RT-qPCR was performed using the Biorad CFX384 Touch System (Biorad Laboratories Inc.). Data was also collected using Biorad CFX96 (Biorad Laboratories Inc.) using 2.5 μM of forward and reverse primer and 2.5 ng of cDNA. Relative mRNA fold expression was calculated using the delta-delta CT method. The Cq value for each treatment was normalized to GAPDH or beta-actin and analyzed in GraphPad Prism (Graph-Pad Software Inc.).

### Transcriptome profiling using Clariom S Array

RNA from cells was run on a Bioanalyzer nano chip (Agilent) for quality control. All samples had a RIN of 9.90 showing excellent RNA quality. cDNA synthesis and array hybridization were prepared using the GeneChip WT Plus Reagent Kit (Affymetrix) according to the manufacturer's recommendations. RNA input was 500 ng per sample. Human Clariom S Array (Applied Biosystems) with a 169 format was used for array hybridization.

### Antibodies

Antibodies used in this study consisted of: PKA-Cα (Santa Cruz Biotechnology, cat. no. sc-28315), GAPDH (Cell Signaling Technologies; cat. no. 5174), Drp1 (BD Biosciences, cat no. 611112), HSP70 (Santa Cruz Biotechnology, cat. no. sc-32239), and Tomm20 (Novus Biologicals, cat. No. NBP2-67501).

## Western blot

Western blot analysis was performed as previously described [44]. Cells were dissociated and lysed in ice-cold RIPA buffer supplemented with protease inhibitors, phosphatase inhibitors, using a Tissue Tearor Homogenizer (Biospec Products, Inc). Whole cell protein lysates (15–30 μg) containing Laemlli buffer plus beta mercaptoethanol, were heated at 95˚C for 10 minutes. The samples were then resolved in 4–20% precast tris-glycine gels (Invitrogen, Carlsbad, CA) and transferred onto PVDF membranes (Bio Rad, Hercules, CA) using Trans-Blot Turbo Transfer System (Bio-Rad, Hercules, CA). Membranes were subjected to blocking for 2 hours at room temperature in 5% non-fat dry milk in tris-buffered saline and 0.1% Tween 20 (TBST). Immunoblot membranes were incubated in primary antibodies at 4˚C overnight. Membranes were then incubated in secondary antibodies at room temperature for 45 minutes followed by three washes with TBST. Immunoblots were visualized using an enhanced chemiluminescence (ECL) kit (Bio-Rad, Hercules, CA and Thermo Scientific, Waltham, MA).

Protein quantification was performed using ImageJ software (normalized to GAPDH). GraphPad Prism 7 (La Jolla, CA) was used to generate graphs and multiple t-test analyses were used to generate p-values.

## Immunofluorescence staining

Immunofluorescence staining was performed as previously described [45]. Cells were grown on coverslips until they reached 60–70% confluence. They were fixed in methanol for 10 minutes at -20˚C, washed three times with DPBS (Dulbecco's Phosphate-Buffered Saline, Thermo Fisher Scientific) and incubated overnight at 4˚C with primary antibody Tomm20 (Novus Biologicals, NBP2-67501). After washing three times with DPBS, the specimens were incubated with Goat anti-Rabbit Alexa Flour 488 secondary antibody (Invitrogen, A-27034) for 1 hour at room temperature. All antibodies were diluted in 0.1% BSA in DPBS. Nuclear staining and mounting was performed with Vectashield mounting medium with DAPI (Vector Laboratories, CA, USA). Imaging was performed using a laser scanning confocal microscope (Zeiss LSM 880). All images were taken at 60x magnification with the same laser setting for comparative analysis between the experimental groups.

## Cell proliferation assay

On Day 0, 25,000 cells were seeded for each cell line into a 24-well plate. Daily cell counts were performed for 6 days in approximate 24-hour time intervals after initial seeding. Media was aspirated from wells and cells were washed in 250uL PBS (calcium and magnesium free; Gibco, 10010023). PBS was aspirated from the wells and 200uL pre-warmed 0.25% trypsin-EDTA was added to the side of the well (Gibco, 25200056), followed by gentle rocking to get complete coverage of the cell layer. Well plates were incubated at 37 degrees C and 5.0% $CO_2$ for approximately 5 minutes or until 90% of cells detached by observation under the microscope. Various volumes of pre-warmed complete media (DMEM, high glucose, Gibco, 11965092; 10% FBS, heat-treated at 56C for 30 minutes, Gibco, 26140079; 5% penicillin-streptomycin 10,000U/mL, Gibco 15140122; 5% Glutamax 100X, Gibco 35050061; 5% sodium pyruvate 100mM, Gibco 11360070) was added to cells, specifically, 200uL on Day 1–3; 600uL on Day 4; and 1200uL on Day 5–6. The medium was dispersed by pipetting over the cell layer several times. 10uL of freshly pipetted cell suspension was added to a cleaned and dried hemocytomer (Bright-Line, Z359629) and observed under the microscope under a 10X objective lens. Cells were counted from the large, central gridded square (1 mm$^2$) and multiplied by 104 to estimate the number of cells per mL, and subsequently multiplied by the total volume of the cell resuspension (mixture of 0.5% trypsin-EDTA and complete media; ranges from 400uL to

1400uL) to calculate the total number of cells per well. Cells were prepared for counting in groups of 8 wells to avoid over-trypsinization and clumping.

### siRNA experiments

Cells were grown to be 70% confluent in 24-well plates prior to transfection with siRNAs. Cells were treated with *PRKACA*-siRNA (assay ID: s11066, Cat. No.: 4390824, Thermo Fisher Scientific) and Silencer Select Negative Control No.1 siRNA (Cat. No.:4390843, Thermo Fisher Scientific) using 1.5μl of siRNA and lipofectamine as delivery vehicle. Lipofectamine RNAiMax reagent (Life Technologies) was used according to the manufacturer's instructions.

### Pharmacologic targeting of PKA and Hsp40

Cells were grown to 70% confluent in 24-well plates prior to treatment with N-[2-[[3-(4-Bromophenyl)-2-propenyl]amino]ethyl]-5-isoquinolinesulfonamide dihydrochloride (H89; R&D Systems) or 3-(1,3-Benzodioxol-5-ylmethylene)-2-oxo-1-pyrrolidinecarboxaldehyde (KNK437, Sigma Aldrich) for 24 hours prior to collection of cells for RNA extraction.

### Transcriptome analysis

Gene expression was analyzed in 23 FLC and 4 non-malignant liver (NML) samples. Of these, 16 FLC and 2 NML samples were published previously [13] (accession: EGAS00001004169). The remaining 7 FLC and 2 NML samples represent a new cohort of samples (accession: GSE181922). Human samples Informed consent was obtained from all human subjects. FLC and non-malignant liver samples were collected from FLC patients according to Institutional Review Board protocols 1802007780, 1811008421 (Cornell University) and/or 33970/1 (Fibrolamellar Cancer Foundation) and provided by the Fibrolamellar Cancer Foundation. As with the published samples, RNA was isolated from the new cohort using the Total RNA Purification Kit (Norgen Biotek). Libraries were prepared using the TruSeq Stranded mRNA Library Prep Kit (Illumina), the KAPA Stranded mRNA-Seq Kit (KAPA Biosystems, Wilmington, MA), or the NEBNext Ultra II Directional Library Prep Kit (New England Biolabs, Ipswich, MA). Sequencing was performed on either the NextSeq500 (Illumina) or the HiSeq2500 (Illumina).

Sequencing results were aligned to the human genome (hg38) using STAR (v2.4.2a) [46] and quantified using Salmon (v0.8) [47]. Differential expression of genes was determined using DESeq2 (v1.29) [48]. Genes with an average expression less than 5 (baseMean) were discarded from the analysis.

### PRKACA immunoprecipitation-mass spectrometry (IP-MS) and data analysis

IP-MS was performed as previously described [49]. Cells were grown to 90% confluency on 150mm dishes, washed with PBS and frozen as cell pellets. Cell pellets from a single dish were used for each replicate and lysed by sonication in lysis buffer [30 mM Tris-HCl, 150 mM NaCl, 0.5% N-dodecylmaltoside, and 1X complete protease and phosphatase inhibitors (Roche). Lysates were cleared by centrifugation and incubated with Dynabeads Protein G (Invitrogen) bound to PKAα antibody (Santa Cruz Biotechnology, sc-903) for 1 hr at 4C. Dynabeads were washed 3X with lysis buffer without detergent, then washed with 100 mM triethylammonium bicarbonate. For protein identification by mass spectrometry, immunoprecipitated proteins were digested directly on Dynabeads by incubation with 1ug Trypsin (Pierce) in 100 mM triethylammonium bicarbonate overnight rotating at 37C. Peptides were

desalted using C18 ZipTip (Millipore) and subjected to reverse-phase LC separation on a 60-min gradient and analyzed on a Q Exactive HF-X (Thermo Fisher Scientific). Data-dependent fragmentation used collision-induced dissociation. RAW files were searched using MaxQuant under standard settings using the UniProt human database, allowing for two missed trypsin cleavage sites, variable modifications for N-terminal acetylation, and methionine oxidation. Candidate peptides and protein identifications were filtered on the basis of a 1% false discovery rate. To examine differences in proteins associated with wild-type PRKACA and the PRKACA-DNAJB1 fusion protein, MaxQuant reported protein groups intensity values were normalized to the intensity value of PRKACA (sp|P17612|KAPCA_HUMAN cAMP-dependent protein kinase catalytic subunit alpha) for each sample. PRKACA normalized intensity values for each protein were log transformed and differential proteins were identified using the limma R/Bioconductor software package.

## Mitochondrial morphometric analysis

Tomm20 z-stacks were background subtracted and automatic threshold values were recorded using Fiji. Tiffs were then uploaded for mitochondrial length and volume quantification using Imaris (Bitplane). Volumetric reconstructions were generated using the Surface tool. Mitochondrial length was measured using the BoundingBox OO Length C setting (longest principal axis of each mitochondria). Mitochondrial volume and length for each image were exported using the export statistics tool. Cumulative frequency of mitochondrial lengths and volumes were determined using GraphPad Prism. Relative frequency of mitochondrial length and volumes were quantified using the 'histogram' tool in Microsoft Excel. Mitochondrial network connectivity measures were determined using Fiji. Specifically, Tomm20 z-stack images were background subtracted and automatic thresholded, after which a binary mask was created. A skeletonization was then created from this binary mask, and the 'Analyze Skeleton' tool was used to determine the number of branches. 'Networks' were defined as any group of mitochondria containing 3 or more branches, while 'large networks' were defined as any group of mitochondria containing 21 or more branches.

## Statistical analysis

Prism software (GraphPad Software, San Diego, CA) was used for statistical analysis.

## Supporting information

**S1 Fig. Full-length blots.** (a) DNA gels used to determine the successful CRISPR deletion. Expected PCR product size for A and B clones was 551bp and for G clones was 615bp. (b) Blots demonstrate the expression of PKA-Cα and DP fusion protein in engineered HEK-DP clones as presented in Fig 1. (c) CREB immunoblot in A9, A11, and HEK-WT performed in triplicate. (d) Phosphorylated CREB in A9, A11, and HEK-WT. (e) α-PKA immunoblot of A9 cells following siRNA treatment (right panel) and corresponding GAPDH immunoblot (left panel).
(PDF)

**S2 Fig. *LINC00473* expression is unchanged in HEK-CT cells compared to HEK-WT cells.** Five HEK-CT clones which were established after transfection with vectors lacking guide RNA sequences have the same level of *LINC00473* expression compared to HEK-WT cells.
(PDF)

**S3 Fig. Hsp70 interacts with PKA-Cα in A9 and A11 cells.** (a) Hsp70 expression is demonstrated in all cell lines. Following co-immunoprecipitation with PKA-Cα antibody, Hsp70 is

identified in the protein complexes from A9 and A11. (b) Full-length blot represented in panel A.
(PDF)

**S4 Fig. BAG2 interacts with DP and PKA-Cα.** (a) Bag2 protein co-immunoprecipitates with
DP and PKA-Cα using PKA antibody. (b) DP and PKA immunoprecipitate using BAG2 anti-
body. (c) Full-length blots represented in a and b panels. GAPDH immunoreactivity is present
in both input blots but not in the immunoprecipitated samples.
(PDF)

**S1 Table. List of potential proteins which interact with DP fusion protein based on proteo-
mic analysis.**
(PDF)

**S2 Table. Gene enrichment pathway analysis based on proteomic data.** Ranked pathways of
significance based on differential protein precipitation in response to PKA-Cα immunoprecip-
itation in A9 and A11 compared to HEK-WT.
(PDF)

## Acknowledgments

Molecular genetics support services were provided by the Boston Children's Hospital Intellec-
tual and Developmental Disabilities Research Center Molecular Genetics Core Facility sup-
ported by U54HD090255 from the NIH Eunice Kennedy Shriver National Institute of Child
Health and Human Development.

The authors extend their gratitude to Dr. Ghazaleh Sadri-Vakili and Dr. Robert Shamberger
for the proofreading and editing of the manuscript.

## Author Contributions

**Conceptualization:** Andrew Emili, Daniel E. Bauer, Praveen Sethupathy, Khashayar Vakili.

**Formal analysis:** Stephanie S. Kim, Ina Kycia, Michael Karski, Rosanna K. Ma, Evan A. Bordt,
Julian Kwan, Anju Karki, Elle Winter, Ranan G. Aktas, Yuxuan Wu, Khashayar Vakili.

**Funding acquisition:** Khashayar Vakili.

**Investigation:** Stephanie S. Kim, Ina Kycia, Michael Karski, Rosanna K. Ma, Evan A. Bordt,
Julian Kwan, Anju Karki, Elle Winter, Ranan G. Aktas, Yuxuan Wu.

**Methodology:** Stephanie S. Kim, Ina Kycia, Michael Karski, Rosanna K. Ma, Evan A. Bordt,
Julian Kwan, Anju Karki, Elle Winter, Ranan G. Aktas, Yuxuan Wu.

**Supervision:** Andrew Emili, Daniel E. Bauer, Praveen Sethupathy, Khashayar Vakili.

**Writing – original draft:** Stephanie S. Kim, Michael Karski, Rosanna K. Ma, Khashayar
Vakili.

**Writing – review & editing:** Stephanie S. Kim, Ina Kycia, Michael Karski, Rosanna K. Ma,
Evan A. Bordt, Julian Kwan, Anju Karki, Elle Winter, Ranan G. Aktas, Yuxuan Wu,
Andrew Emili, Daniel E. Bauer, Praveen Sethupathy, Khashayar Vakili.

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
