## [Decision Letter · Decision Letter 0]

29 Sep 2021

PONE-D-21-26897DNAJB1-PRKACA in HEK293T cells induces LINC00473 overexpression that depends on PKA signalingPLOS ONE

Dear Dr. Vakili,

Thank you for submitting your manuscript to PLOS ONE. After careful consideration, we feel that it has merit but does not fully meet PLOS ONE’s publication criteria as it currently stands. Therefore, we invite you to submit a revised version of the manuscript that addresses the points raised during the review process. Please respond to all critique, point-by-point. In particular: Authors should compare their manipulated clonal lines with irrelevantly manipulated lines instead of unmanipulated bulk cells. Moreover, in Fig. 3 they should provide statistics. A validation of the mass spectrometry data is missing in Fig.7. 

We look forward to receiving your revised manuscript.

Kind regards,

Klaus Roemer

Academic Editor

PLOS ONE

Journal Requirements:

2. In your Methods section, please provide additional details regarding the cell lines used in your study and ensure you have described the source. For more information regarding PLOS' policy on materials sharing and reporting, see https://journals.plos.org/plosone/s/materials-and-software-sharing#loc-sharing-materials, and for more information on PLOS ONE's guidelines for research using cell lines, see https://journals.plos.org/plosone/s/submission-guidelines#loc-cell-lines.

3. Please provide additional details regarding participant consent. In the ethics statement in the Methods and online submission information, please ensure that you have specified (1) whether consent was informed and (2) what type you obtained (for instance, written or verbal, and if verbal, how it was documented and witnessed). If your study included minors, state whether you obtained consent from parents or guardians.

Reviewers' comments:

Reviewer's Responses to Questions

**Comments to the Author**

1. Is the manuscript technically sound, and do the data support the conclusions?

Reviewer #1: No

Reviewer #2: Partly

2. Has the statistical analysis been performed appropriately and rigorously? 

Reviewer #1: No

Reviewer #2: N/A

3. Have the authors made all data underlying the findings in their manuscript fully available?

Reviewer #1: No

Reviewer #2: Yes

4. Is the manuscript presented in an intelligible fashion and written in standard English?

Reviewer #1: Yes

Reviewer #2: Yes

5. Review Comments to the Author

Reviewer #1: General Comments:

In this manuscript, Kim et al. describe the generation and initial characterization of a HEK 293T-derived cell line carrying a large deletion to mimic the naturally occurring DNAJB1-PRKACA fusion gene, a driver mutation of fibrolamellar carcinoma (FLC). They use CRISPR-Cas9 to introduce this deletion and confirm its existence in twelve separate clonal lines, including both hetero- and homozygous genotypes. The authors choose two of those clonal lines and compare them to the bulk, parental cell line with a variety of descriptive methods, including gene expression by microarray and RT-qPCR; proliferation assay; mass spectrometry; and fluorescence microscopy of mitochondrial markers. Some of the results mirror changes known to occur in FLC, and thus the authors conclude that their cell lines are a suitable model for the study of this carcinoma.

While the underlying idea to generate these cell lines has merit, the study design is fundamentally flawed and most of the conclusions are therefore not supported by the data.

Major Concerns:

1) Throughout this manuscript, the authors directly compare two monoclonal cell lines, which have undergone transfection, FACS and single-cell selection, to an apparently completely untreated bulk population that has not undergone any of these procedures. That this is highly problematic is evident in the authors own data, as the variability between the individual clones, where examined, is extensive (e.g., Fig. 2, Fig. 3, Fig. 7b). An appropriate experimental setup must include several wildtype clones that have undergone the same treatment, including transfection with CRISPR-Cas9 plasmids that do not carry a sgRNA sequence. As these controls were not included, the vast majority of the experiments reported here are not sound and unfortunately cannot be used to draw conclusions.

2) For their microarray analysis, the authors do not use replicates. While suboptimal, this can be done but should at least be evaluated with a more stringent fold-change cut-off. Typically, genes at the lower end of expression fluctuate considerably in microarray analyses and should be excluded from analysis, which the authors apparently neglected to do. To provide an assessment of this variability, the authors should show an MA plot.

3) The authors provide no statistical analysis of the data presented in Fig. 3.

4) In Fig. 4 and 5, the authors do not include wildtype controls at all. It would also have been interesting to see if H89 treatment affects the levels of CGA.

5) The difference in proliferation between wildtype and the deletion clones is wholly unconvincing, in particular in the absence of proper wildtype controls.

6) There is no validation of the mass spectrometry results shown in Fig. 7.

7) The data on mitochondrial fission would be considerably more convincing had the authors shown some sort of rescue experiment, e.g. with a knockdown of the DNAJB1-PRKACA gene.

Minor Concerns:

8) The authors should provide the PCR results confirming the deletion in their initial screen and provide the primer sequences used.

9) The authors should provide accession numbers for the old and new RNA-seq data derived from FLC patients and controls used in Fig. 2d.

10) The authors should provide some sort of gene-enrichment analysis for their microarray data (e.g., gene ontology, as done for the mass spectrometry data, and/or GSEA).

11) In Fig. 4, the authors used the unpaired t-test, although their reference sample has a standard deviation of 0. This violates one of the underlying assumptions of the paired and unpaired t-tests, namely that the means of both groups being compared follow normal distributions. The appropriate test for this situation is a one-sample t-test, using the mean of the reference group.

12) In Fig. 5, is there any reason for the dramatic difference in responsiveness to H89 treatment between the two transcript variants of LINC00473?

Reviewer #2: Kim et al. developed and characterized a new cellular model (HEK-DP) that can be used to study Fibrolamellar(FLP) carcinoma as an alternative to the animal model. FLP is a rare and aggressive liver cancer that predominantly affects adolescents and young adults. The primary oncogenic driver originates from a ~400 kb deletion in chromosome 19 that generated an in-frame fusion of exon 1 from the DNAJB1 gene with exons 2-10 of the PRKACA gene. The chimeric kinase (DP) is fully functional, but the fusion gene's mechanisms that lead to FLC are still unclear.

Here are some points that I think the authors should address:

• Figure 2 (a-b): it is hard to read the name of the upregulated/underregulated gene:

• In figure 3 shows that the expression of CGA and LINC00473 is upregulated in the majority of HEK-DP. In the main text, the authors said:

“Subsequent quantitative analysis of the gene chip findings using RT-qPCR confirmed the significant increase in CGA and LINC00473 in all 12 HEK-DP clones (Figure 3).”

Based on this, why is the upregulation of these genes not consistent within all the clones? Can the authors comment on this?

• Figure 4: It is hard to read the text, so it is unclear what control the authors have used to perform the siRNA targeting PRKACA analysis.

• In figure 5, the authors reported the pharmacologic targeting of PKA with H89 and Hsp40 with KBK437. Can the author clarify why they used those concentrations of inhibitors? Can they also express those concentrations as molar-ratio instead of molarity? Why does LINC00473(TV2) expression level increase at lower concentrations of H89?

Overall, the authors have developed a potential alternative cell line to study FLP in a more controlled system. However, we believe that more cell line characterization needs to be done to understand if the non-liver cell line can be used as an alternative to the animal model. Nevertheless, we understand that this is preliminary work and, with the appropriate clarification, it should be accepted by your journal.

6. PLOS authors have the option to publish the peer review history of their article (what does this mean?). If published, this will include your full peer review and any attached files.

Reviewer #1: No

Reviewer #2: No

---

## [Author Response · Author response to Decision Letter 0]

18 Jan 2022

PONE-D-21-26897

DNAJB1-PRKACA in HEK293T cells induces LINC00473 overexpression that depends on PKA signaling

PLOS ONE

Response to Reviewers

Reviewer #1: General Comments:

In this manuscript, Kim et al. describe the generation and initial characterization of a HEK 293T-derived cell line carrying a large deletion to mimic the naturally occurring DNAJB1-PRKACA fusion gene, a driver mutation of fibrolamellar carcinoma (FLC). They use CRISPR-Cas9 to introduce this deletion and confirm its existence in twelve separate clonal lines, including both hetero- and homozygous genotypes. The authors choose two of those clonal lines and compare them to the bulk, parental cell line with a variety of descriptive methods, including gene expression by microarray and RT-qPCR; proliferation assay; mass spectrometry; and fluorescence microscopy of mitochondrial markers. Some of the results mirror changes known to occur in FLC, and thus the authors conclude that their cell lines are a suitable model for the study of this carcinoma.

Response: We appreciate the suggestions provided by Reviewer #1, which have further strengthen our manuscript. However, we DO NOT claim that our model is “a suitable model for the study of this carcinoma”. In fact, in the discussion section of the manuscript we had clearly stated “One of the limitations of our cell model is that it is not a suitable model to study the oncogenic mechanism of DP in liver cells since the HEK293 cells are not of the same origin and at baseline have a different gene expression pattern.” 

We also had stated the following in the Discussion section: “Therefore, our engineered HEK-DP cell lines may have the potential to provide a platform for understanding the regulatory link between DP and LINC00473 expression, which is present in FLC tumors, and provide the field with a cellular system for screening DP-specific inhibitors by using LINC00473 as a candidate marker for DP activity.” 

While the underlying idea to generate these cell lines has merit, the study design is fundamentally flawed and most of the conclusions are therefore not supported by the data.

Major Concerns:

1) Throughout this manuscript, the authors directly compare two monoclonal cell lines, which have undergone transfection, FACS and single-cell selection, to an apparently completely untreated bulk population that has not undergone any of these procedures. That this is highly problematic is evident in the authors own data, as the variability between the individual clones, where examined, is extensive (e.g., Fig. 2, Fig. 3, Fig. 7b). An appropriate experimental setup must include several wildtype clones that have undergone the same treatment, including transfection with CRISPR-Cas9 plasmids that do not carry a sgRNA sequence. As these controls were not included, the vast majority of the experiments reported here are not sound and unfortunately cannot be used to draw conclusions.

Response: We appreciate the input. We have subsequently developed 5 clones from wild-type HEK293 cells which have undergone transfection with vectors lacking guide RNA sequences used for construction of HEK-DP cells, FACS, and single-cell selection. These clones did not demonstrate an increase in LINC00473 expression compared to wild-type cells. Results have been added to the supplementary data. 

2) For their microarray analysis, the authors do not use replicates. While suboptimal, this can be done but should at least be evaluated with a more stringent fold-change cut-off. Typically, genes at the lower end of expression fluctuate considerably in microarray analyses and should be excluded from analysis, which the authors apparently neglected to do. To provide an assessment of this variability, the authors should show an MA plot.

Response: The purpose of the microarray was to perform a cursory examination of the transcriptome in HEK-DP cells. In Figure 2, we have only included genes with greater than 2-fold change. We have not performed any GO analysis or proposed any pathways of significance since we do not think this is directly relevant to FLC biology. However, through the microarray and subsequent qPCR at Vakili lab and independently at Sethupathy lab, we identified LINC00473 as an upregulated transcript leading us to pursue whether this could be a marker for DP activity. Even if we set our fold-change at a higher cut-off of 5, we would still identify LINC00473 as a significantly upregulated transcript. 

3) The authors provide no statistical analysis of the data presented in Fig. 3.

Response: Since the focus of our work is on clones A9 and A11, we have revised the figure. The updated figure demonstrates statistical LINC00473 expression in A9 and A11.

4) In Fig. 4 and 5, the authors do not include wildtype controls at all. It would also have been interesting to see if H89 treatment affects the levels of CGA.

Response: Since there is a 10-fold difference in LINC00473 expression between wildtype HEK cells and the HEK-DP clones, we did not feel it would be relevant to include wildtype cells for these experiments. 

5) The difference in proliferation between wildtype and the deletion clones is wholly unconvincing, in particular in the absence of proper wildtype controls.

Response: We agree that the difference is not robust in the first 3 days, however, statistically significant after 3 days in A11 cells. Our hypothesis is NOT that the presence of DP will increase proliferation. Therefore, we did not perform extensive experiments to measure proliferation rates of the clones in response to DP targeting due to small difference in the rates of proliferation between HEK-DP and HEK-WT cells. We do hypothesize that once a cell transforms into a cancer cell, then the proliferation rate changes significantly. However, our HEK-DP cells are not considered cancer cells and it may take several generations of the clones and continued exposure to DP before they start demonstrating significant alterations in their proliferation rates. We simply included this graph as we expected that reviewers will ask about the proliferation rates of the DP clones. 

6) There is no validation of the mass spectrometry results shown in Fig. 7.

Response: We have now included a validation experiment in order to assess the interaction of BAG2 and DP based on the proteomic data. Please see Supplementary Figure S3. The co-IP experiments demonstrate interaction between BAG2 and DP as well as between BAG2 and PKA-C� Based on the proteomic data and the intensity of the bands on the blots, the affinity of DP to form a complex with BAG2 may be higher than PKA-C�. We will study this interaction in more detail in the future. We have outlined the potential importance of DP-BAG2-HSP70 interaction in the Discussion section of the manuscript.

7) The data on mitochondrial fission would be considerably more convincing had the authors shown some sort of rescue experiment, e.g. with a knockdown of the DNAJB1-PRKACA gene.

Response: We agree that this experiment would be of great value and we plan to pursue this in the near future. We are aware of a DNAJB1-PRKACA-specific siRNA that is under optimization. Once developed, it would provide the best strategy to perform the rescue experiments. 

Minor Concerns:

8) The authors should provide the PCR results confirming the deletion in their initial screen and provide the primer sequences used.

Response: The DNA gels confirming the deletion have been added to Supplementary Figure S1 and the primer sequences added to the Methods section. 

9) The authors should provide accession numbers for the old and new RNA-seq data derived from FLC patients and controls used in Fig. 2d.

Response: The accession numbers have been added to the Methods section. 

10) The authors should provide some sort of gene-enrichment analysis for their microarray data (e.g., gene ontology, as done for the mass spectrometry data, and/or GSEA).

Response: See #2.

11) In Fig. 4, the authors used the unpaired t-test, although their reference sample has a standard deviation of 0. This violates one of the underlying assumptions of the paired and unpaired t-tests, namely that the means of both groups being compared follow normal distributions. The appropriate test for this situation is a one-sample t-test, using the mean of the reference group.

Response: Figure 4 has been updated and includes the raw mean and SD of the reference group.

12) In Fig. 5, is there any reason for the dramatic difference in responsiveness to H89 treatment between the two transcript variants of LINC00473?

Response: The difference may be due to the differential sensitivity of the 2 transcript variants to PKA activity/inhibition. 

Reviewer #2: Kim et al. developed and characterized a new cellular model (HEK-DP) that can be used to study Fibrolamellar(FLP) carcinoma as an alternative to the animal model. FLP is a rare and aggressive liver cancer that predominantly affects adolescents and young adults. The primary oncogenic driver originates from a ~400 kb deletion in chromosome 19 that generated an in-frame fusion of exon 1 from the DNAJB1 gene with exons 2-10 of the PRKACA gene. The chimeric kinase (DP) is fully functional, but the fusion gene's mechanisms that lead to FLC are still unclear.

Response: We appreciate the input and suggestions provided by Reviewer #2, which will strengthen our manuscript.

Here are some points that I think the authors should address:

• Figure 2 (a-b): it is hard to read the name of the upregulated/underregulated gene:

Response: Figure 2 has been revised to improve readability.

• In figure 3 shows that the expression of CGA and LINC00473 is upregulated in the majority of HEK-DP. In the main text, the authors said:

“Subsequent quantitative analysis of the gene chip findings using RT-qPCR confirmed the significant increase in CGA and LINC00473 in all 12 HEK-DP clones (Figure 3).”

Based on this, why is the upregulation of these genes not consistent within all the clones? Can the authors comment on this?

Response: Since the clones were established from single cells following FACS, it is likely that there is some clonal variability, which is an inherent issue with cell lines. However, the important finding is that there is upregulation of CGA and LINC00473 in the majority of the clones expressing DP. Since we have used 2 clones for our analyses, we have simplified this figure to improve clarity. 

• Figure 4: It is hard to read the text, so it is unclear what control the authors have used to perform the siRNA targeting PRKACA analysis.

Response: We have revised the figure by increasing its size. The control is the untreated A9 cell line.

• In figure 5, the authors reported the pharmacologic targeting of PKA with H89 and Hsp40 with KBK437. Can the author clarify why they used those concentrations of inhibitors? Can they also express those concentrations as molar-ratio instead of molarity? Why does LINC00473(TV2) expression level increase at lower concentrations of H89?

Response: The concentrations were based on previously published studies, which were cited in the manuscript. We used molar concentration since it is a widely used unit for in vitro drug effect experiments. It appears that LINC00473 TV1 expression is more sensitive to PKA inhibition by H89 compared to TV2. There is no statistically significant increase in TV2 expression at lower concentrations of H89. 

Overall, the authors have developed a potential alternative cell line to study FLP in a more controlled system. However, we believe that more cell line characterization needs to be done to understand if the non-liver cell line can be used as an alternative to the animal model. Nevertheless, we understand that this is preliminary work and, with the appropriate clarification, it should be accepted by your journal.

Response: Thank you. We agree with the stated sentiments. Our goal is not to use this model as an alternative to an animal model or as a model for FLC oncogenesis. Since we see an overexpression of LINC00473 in FLC and we have established a clear link between LINC00473 and DP in our model, this model may provide a tool for elucidating their connection, which may provide some insights relevant to FLC. In a field with limited cell models, we believe our model will contribute positively to the field.

---

## [Editor Report · Decision Letter 1]

28 Jan 2022

DNAJB1-PRKACA in HEK293T cells induces LINC00473 overexpression that depends on PKA signaling

PONE-D-21-26897R1

Dear Dr. Vakili,

We’re pleased to inform you that your manuscript has been judged scientifically suitable for publication and will be formally accepted for publication once it meets all outstanding technical requirements.

Kind regards,

Klaus Roemer

Academic Editor

PLOS ONE
---

## [Editor Report · Acceptance letter]

4 Feb 2022

PONE-D-21-26897R1 

*DNAJB1-PRKACA* in HEK293T cells induces *LINC00473* overexpression that depends on PKA signaling 

Dear Dr. Vakili:

I'm pleased to inform you that your manuscript has been deemed suitable for publication in PLOS ONE. Congratulations! Your manuscript is now with our production department. 

Kind regards, 

on behalf of

Dr. Klaus Roemer 

Academic Editor

PLOS ONE